# A MIMO Radar Signal Processing Algorithm for Identifying Chipless RFID Tags

**DOI:** 10.3390/s21248314

**Published:** 2021-12-12

**Authors:** Chen Su, Chuanyun Zou, Liangyu Jiao, Qianglin Zhang

**Affiliations:** School of Information Engineering, Southwest University of Science and Technology, Mianyang 621010, China; zoucy@swust.edu.cn (C.Z.); jiaoly@swust.edu.cn (L.J.); zhangql@mails.swust.edu.cn (Q.Z.)

**Keywords:** chipless radio-frequency identification (RFID) tag, multi-tag identification, MIMO algorithm, azimuth estimation

## Abstract

In this paper, the multiple-input, multiple-output (MIMO) radar signal processing algorithm is efficiently employed as an anticollision methodology for the identification of multiple chipless radio-frequency identification (RFID) tags. Tag-identifying methods for conventional chipped RFID tags rely mostly on the processing capabilities of application-specific integrated circuits (ASICs). In cases where more than one chipless tag exists in the same area, traditional methods are not sufficient to successfully read and distinguish the IDs, while the direction of each chipless tag can be obtained by applying MIMO technology to the backscattering signal. In order to read the IDs of the tags, beamforming is used to change the main beam direction of the antenna array and to receive the tag backscattered signal. On this basis, the RCS of the tags can be retrieved, and associated IDs can be identified. In the simulation, two tags with different IDs were placed away from each other. The IDs of the tags were successfully identified using the presented algorithm. The simulation result shows that tags with a distance of 0.88 m in azimuth can be read by a MIMO reader with eight antennas from 3 m away.

## 1. Introduction

Radiofrequency identification (RFID) technology is a wireless data capture technology. This technology uses radio waves to exchange information between the reader and radio frequency tag, and it realizes automatic object recognition. RFID has the advantages of being waterproof and antimagnetic as well as having high temperature resistance, large data storage, long service life, and high efficiency; it has been widely used in logistics, warehousing, identification, asset management, and other aspects. RFID tags are widely deployed in the Internet of Things system, and the tag cost has become an important factor restricting the application of RFID technology. Manufacturing low-cost RFID tags for largescale commercial use is the primary goal of RFID technology development. Recently, many efforts have been made to improve the design of printable and compact chipless RFID tags [1,2,3,4,5] and identification techniques [6,7,8]. Chipless RFID tags are similar to ordinary bar codes and QR codes in function, but their applications are wider. Their advantages include recognition in dark environments, lower cost for mass production, potential for integration with green technologies, and potential for conversion to sensors [9]. Chipless RFID tags do not need any digital chips, so the corresponding digital communication conversion protocol is simplified. This makes the system design simpler and more convenient.

In tag-intensive application scenarios, there are often two or more tags within the scope of a reader. When these tags respond to the query command issued by the reader at the same time, the multi-tag response signals are mixed together, causing the reader to be unable to distinguish them. This phenomenon is called multi-tag conflict [10]. At present, the RFID anticollision algorithm mainly includes four methods: frequency division multiple access (FDMA), time division multiple access (TDMA), code division multiple access (CDMA), and space division multiple access (SDMA) [11]. Due to the chipless RFID tag having no communication protocol or only a simple MAC layer protocol, the first three methods are generally not applicable. Although the MAC layer communication protocol proposed in the literature [12] can be used for multi-tag identification, it occupies bandwidth and reduces information capacity. Multiple chipless tag identification has not yet been thoroughly investigated. SDMA divides the communication channel into several subchannels according to the space position, and the simultaneous signals at the same frequency can be distinguished by the difference in the propagation path. The core technology of SDMA is smart antenna technology. Most of the published work using SDMA technology focuses on anti-collision of chipped RFID tags. In these studies, multi-antenna technology was used to distinguish chipped RFID tags. With the help of space characteristics and chip processing capability, the anti-collision method proved to be effective under certain conditions.

The reader designed by Abderrzazk in the literature [13] uses multiple antennas installed at different locations to divide the reading zone into different subsets. By selecting different antennas and different tag subsets, the anti-collision capability of a multi-tag system is improved. However, this scheme is only suitable for the application scenarios with wide tag distribution range, and it is unable to realize anti-collision for tags in the same subset. To achieve tag anti-collision in a smaller area, some works have been carried out based on the transmission delay of signals to antennas at different positions. In the literature [14], Zhongqi Liu proposed an anti-collision algorithm based on the different communication times generated by different distances between the tag and the reader antennas. Zhu proposed a method to identify multiple surface acoustic wave tags by calculating the direction of arrival of the signal [15]. In the literature [16], Jiexiao Yu realized space division recognition of UHF tags by using secondary digital beamforming technology. In Colby Boyer’s work [17], the channel characteristics of multiple-input, multiple-output (MIMO) RFID were analyzed, and the characteristics of the secondary backscattering channel were summarized. The above work mainly focused on anticollision analysis of passive RFID chipped tags in the UHF frequency band, but did not study the chipless tags. In addition, a relatively simple beamforming method was used without the use of MIMO signal processing.

In this paper, based on the read principle of frequency-domain-coding chipless RFID tag, a multi-tag recognition technology based on MIMO is proposed. MIMO signal processing technology is used to realize the identification of multiple chipless RFID tags by means of linear frequency-modulated continuous waves. In the following, we describe the basic theory in Section 2, and a chipless tag reading model based on MIMO is proposed in Section 3. The performance analysis and simulation are presented in Section 4. Conclusions are given in Section 5.

## 2. Theoretical Basis

Figure 1 shows the basic principle of RFID. According to the principle of tag reading, chipless RFID tags can be divided into five categories: tags based on image shape [18], tags based on time domain response [19], tags based on frequency domain response [20], tags based on phase/amplitude modulation [21], and mixed coding tags. The frequency-domain tag [22,23,24] has a large data capacity and can be fully printed at low cost, making it the most promising technology to achieve commercialization and marketization. It has been the key direction of chipless RFID tag design research.

The frequency-domain chipless RFID tag uses a radar scattering structure to encode information. The reader radiates a wideband signal to the tag, resulting in electromagnetic scattering. The scattered electromagnetic wave is composed of two parts. One part is reflected by the tag surface due to the difference in wave impedance characteristics between the tag and the air. The other part is caused by the discontinuous surface of the tag, whereby the electromagnetic wave is diffracted in these places. From the perspective of induced current, the scattering field is formed by the induced electromagnetic current and the secondary radiation of electromagnetic charge generated by the incident electromagnetic field on the tag surface. Obviously, the radiation field is related to the structure of the tag and contains the tag information.

The tag’s information can be described by the radar scattering cross section (RCS). The RCS is often used to describe the scattering ability of an object in a specified direction; it is defined as the ratio of the total scattering power $P_{s}$ of the scatterer in the specified direction to the power density $w_{i}$ of the electromagnetic wave incident on the scatterer surface. The RCS can be written as:(1)$$RCS=\frac{P_{s}}{w_{i}}=4\pi R^{2}\frac{E_{s}^{2}}{E_{i}^{2}}$$
where $E_{s}$ and $E_{i}$ are the electric field scattered by the scatterer at distance R and the electric field incident on the scatterer by the corresponding field source, respectively. For a traditional chipped tag, the $RCS_{\mathrm{tag}}$ can be written as [25]:(2)$$RCS_{\mathrm{tag}}=\frac{\lambda^{2}R_{a}G_{tag}^{2}}{\pi \left|Z_{a}+Z_{c}\right|}$$
where λ represents the wavelength of the radio frequency electromagnetic wave, $Z_{a}$ is the input impedance of the antenna on the tag, and $R_{a}$ represents the real part of $Z_{a}$, $Z_{c}$ is the equivalent impedance of the silicon chip, and $G_{tag}$ represents the gain of the antenna. For a chipless RFID tag, the tag is equivalent to an antenna. $Z_{a}$ and $Z_{c}$ can be combined into one impedance value. When the tag is in a state of resonance, the input impedance $Z_{a}$ is a real number, namely, $\left|Z_{a}\right|=R_{a}$. Thus, the RCS of a chipless RFID tag can be written as:(3)$$RCS_{\mathrm{tag}}=\frac{\lambda^{2}R_{a}G_{tag}^{2}}{\pi \left|2Z_{a}\right|}=\frac{\lambda^{2}G_{tag}^{2}}{2\pi }$$

The chipless RFID tag can be considered equivalent to a radar target when it works. The transmission relationship of electromagnetic energy between the reader and the tag can be expressed by the radar ranging equation. The relationship between the receiving and transmitting power of the reader can be expressed as:(4)$$\frac{P_{r}}{P_{t}}=RCS_{tag}\frac{G_{r}G_{t}}{4\pi }{\left[\frac{\lambda }{4\pi R^{2}}\right]}^{2}$$
where $P_{r}$ and $P_{t}$ are the receiving and transmitting power of the reader antenna, respectively, and $G_{r}$ and $G_{t}$ are the receiving and transmitting gain of the antenna, respectively. Equation (3) can then be rewritten as:(5)$$RCS_{tag}=k\frac{P_{r}}{P_{t}}$$
in which:(6)$$k=\frac{{\left(4\pi \right)}^{3}{\left(R_{1}R_{2}\right)}^{2}}{\lambda^{2}G_{r}G_{t}}$$

By measuring the power of the received signal and compensating the gain of the antenna, the reader can obtain the RCS of the chipless RFID tag.

In order to improve the coding capacity of chipless tags, frequency shift keying (FSK) and variable polarization coding were introduced on the basis of frequency-domain amplitude coding (OOK). For example, Professor Jalaly proposed the dipole array chipless tag [26]. Professor Vena proposed the ring-structure chipless tag in the literature [27]. In 2016, scholars from the South China University of Technology improved the structure of the square ring resonator [28]. In reference [29], a printable label structure was proposed, composed of two groups of symmetrical right-angle resonators. These chipless RFID tags are shown in Figure 2.

## 3. Algorithm

Because the frequency-domain-encoding chipless tag mentioned in Section 2 requires a wide bandwidth, LFCW is used in the novel identifying algorithm to obtain the tag RCS. With the help of MIMO radar signal processing, the chipless RFID tag azimuth location is acquired. On this basis, the signal-to-noise ratio (SNR) of the scattered signal is improved by beamforming technology, and the tag is read.

The MIMO radar adopted in this algorithm has *M*-many transmitting antennas and *N*-many receiving antennas. For the convenience of analysis, it is assumed that *M* = *N* and that the transmitting and receiving antenna positions are the same.

### 3.1. Received Signal

As Figure 3 shows, the antenna array is a uniform linear array, and the distance between adjacent antennas is d. Let each antenna-transmitted signal $s_{1}\left(t\right)$, $s_{2}\left(t\right),\ldots ,s_{M}\left(t\right)$ be a linear frequency-modulated continuous wave:(7)$$s_{i}\left(t\right)=e^{-j2\pi f_{i}t-j\pi kt^{2}}-\frac{T_{p}}{2}\leq t\leq \frac{T_{p}}{2}$$

The wideband signal characteristics are used to identify chipless RFID tags encoded in the frequency domain. $f_{i}$ is the center frequency of the ith wave, k is the frequency modulation rate, and $T_{p}$ is the transmitted signal pulse width. Emitted signals are transmitted through space, and the synthesized signal at the chipless RFID tag located at distance R and azimuth φ is as follows:(8)$$p\left(t\right)=\xi_{1}\overset{M}{\underset{i=1}{\sum }}s_{i}\left(t-\tau_{i}-\tau_{r}/2\right)$$
where $\xi_{1}$ is the propagation attenuation factor, assumed to be the same for all signals, $\tau_{r}=2R/c$ is the two-way propagation delay from each antenna to the tag, and $\tau_{i}$ is the signal delay difference between the ith (i=1,2,⋯,M) antenna and the reference antenna to the tag. When the MIMO array is a linear array, it can be expressed as:(9)$$\tau_{i}=\frac{\left(i-1\right)d\sin\theta }{c}$$
in which θ is the angle between the target line and the antenna array and c is the speed of light. Then, p(t) can be written as:(10)$$p\left(t\right)=\xi_{1}\sum_{i=1}^{M}s_{i}\left(t-\tau_{r}/2\right)e^{-j\left(i-1\right)\varphi_{f}}$$ To simplify the calculation, $\varphi_{f}$ is approximated as:(11)$$\varphi_{f}=\frac{2\pi d\sin\theta }{c/f_{c}}$$
where $f_{c}$ is the center frequency of the system. Signal p(t) is reflected by the chipless RFID tag, and the echo signal received by the mth antenna is:(12)$$x_{m}\left(t\right)=\xi_{2}\xi_{1}e^{-j\left(m-1\right)\varphi_{f}}\overset{M}{\underset{i=1}{\sum }}s_{i}\left(t-\tau_{r}\right)e^{-j\left(i-1\right)\varphi_{f}}+v_{m}\left(t\right)$$
where $v_{m}\left(t\right)$ is the noise received by the mth antenna and $\xi_{2}$ is the receiving propagation attenuation factor. The signals received by the array are written in matrix form as:(13)$$X\left(t\right)=\xi a\left(\varphi \right)a^{T}\left(\varphi \right)S\left(t-\tau_{r}\right)+V\left(t\right)$$
where:(14)$$\xi =\xi_{2}\xi_{1}$$
(15)$$a\left(\varphi \right)=\left[\begin{array}{c} 1\\ e^{-j\varphi_{f}}\\ \begin{array}{c} \vdots \\ e^{-j\left(M-1\right)\varphi_{f}} \end{array} \end{array}\right]$$
(16)$$S\left(t-\tau_{r}\right)=\left[\begin{array}{c} s_{1}\left(t-\tau_{r}\right)\\ s_{2}\left(t-\tau_{r}\right)\\ \begin{array}{c} \vdots \\ s_{M}\left(t-\tau_{r}\right) \end{array} \end{array}\right]$$
(17)$$V\left(t\right)=\left[\begin{array}{c} v_{1}\left(t\right)\\ v_{2}\left(t\right)\\ \begin{array}{c} \vdots \\ v_{M}\left(t\right) \end{array} \end{array}\right]$$

### 3.2. MIMO Signal Processing

MIMO signal processing mainly consists of transmitting beamforming and receiving beamforming [30]. When the estimated direction is consistent with the actual direction of the tag, the output signal power reaches the maximum. Let $y_{b}\left(t\right)$ be the received beamforming result in the estimated direction $\varphi_{b}$:(18)$$y_{b}\left(t\right)=a^{H}\left(\varphi_{b}\right)x\left(t\right)=\xi \left\{\frac{\sin\left[\frac{M}{2}\left(\varphi_{f}-\varphi_{b}\right)\right]}{\sin\left[\frac{1}{2}\left(\varphi_{f}-\varphi_{b}\right)\right]}e^{j\frac{M-1}{2}\left(\varphi_{f}-\varphi_{b}\right)}\right\}\overset{M}{\underset{i=1}{\sum }}s_{i}\left(t-\tau_{r}\right)e^{-j\left(i-1\right)\varphi_{f}}+\overset{M}{\underset{i=1}{\sum }}v_{i}\left(t\right)e^{-j\left(i-1\right)\varphi_{b}}$$
where $a^{H}\left(\varphi_{b}\right)$ is the conjugate matrix of a(φ) and $y_{b}\left(t\right)$ passes the band-pass filters with center frequency $f_{i}$ and bandwidth $kT_{p}$. The signal of each antenna working band frequency is as follows:(19)$$y_{bm}\left(t\right)=\xi \left\{\frac{\sin\left[\frac{M}{2}\left(\varphi_{f}-\varphi_{b}\right)\right]}{\sin\left[\frac{1}{2}\left(\varphi_{f}-\varphi_{b}\right)\right]}e^{j\frac{M-1}{2}\left(\varphi_{f}-\varphi_{b}\right)}\right\}s_{m}\left(t-\tau_{r}\right)e^{-j\left(m-1\right)\varphi_{f}}+v_{bm}\left(t\right)$$
where $v_{bm}\left(t\right)$ is the noise generated by the noise in Equation (18) after the bandpass filter. The output $y_{b1},y_{b2},\cdots ,y_{bM}$ is written in vector form as:(20)$$y_{b}\left(t\right)=\xi \left\{\frac{\sin\left[\frac{M}{2}\left(\varphi_{f}-\varphi_{b}\right)\right]}{\sin\left[\frac{1}{2}\left(\varphi_{f}-\varphi_{b}\right)\right]}e^{j\frac{M-1}{2}\left(\varphi_{f}-\varphi_{b}\right)}s_{m}\left(t-\tau_{r}\right)\right\}\;a\left(\varphi \right)+V_{b}\left(t\right)$$
where:(21)$$V_{b}\left(t\right)=\left[\begin{array}{c} v_{b1}\left(t\right)\\ v_{b2}\left(t\right)\\ \begin{array}{c} \vdots \\ v_{bM}\left(t\right) \end{array} \end{array}\right]$$

Transmitting beamforming is carried out in the $\varphi_{b}$ direction, and the output obtained is as follows:(22)$$y\left(t\right)=a^{H}\left(\varphi_{b}\right)y_{b}\left(t\right)=\xi \left\{\frac{\sin\left[\frac{M}{2}\left(\varphi_{f}-\varphi_{b}\right)\right]}{\sin\left[\frac{1}{2}\left(\varphi_{f}-\varphi_{b}\right)\right]}e^{j\frac{M-1}{2}\left(\varphi_{f}-\varphi_{b}\right)}\right\}\left\{\frac{\sin\left[\frac{M}{2}\left(\varphi_{f}-\varphi_{b}\right)\right]}{\sin\left[\frac{1}{2}\left(\varphi_{f}-\varphi_{b}\right)\right]}e^{j\frac{M-1}{2}\left(\varphi_{f}-\varphi_{b}\right)}\right\}\overset{M}{\underset{m=1}{\sum }}s_{m}\left(t-\tau_{r}\right)+\overset{M}{\underset{m=1}{\sum }}v_{bm}e^{j\left(m-1\right)\varphi_{b}}$$

### 3.3. Azimuth Estimation

Obviously, $\varphi_{b}$ is related to the energy of signal y(t) in Equation (22). The energy of signal y(t) in Equation (22) reaches the maximum value when y(t) is in the same direction as the actual direction of the tag. The azimuth estimation $\varphi^{\prime }$ can be expressed as:(23)$$\varphi^{\prime }=\arg\underset{-\frac{\pi }{2}\leq \varphi_{b}\leq \frac{\pi }{2}}{\max}\int_{-\infty }^{+\infty }{\left|y\left(t\right)\right|}^{2}dt$$

Traversing $\varphi_{b}$ from 0 to 180 degrees, a direction diagram can be drawn. The maximum point is the azimuth direction of the tag.

### 3.4. Algorithm Flowchart

A flowchart of the proposed algorithm is illustrated in Figure 4. The aforementioned algorithm is general and can be used for any frequency-domain amplitude-coding chipless tag reading.

By substituting the $\varphi^{\prime }$ obtained by azimuth estimation into MIMO signal processing, spatial filtering can be realized. The received signal after MIMO processing is as follows:(24)$$y_{tag}\left(t\right)=y\left(t\right){|}_{\varphi_{b}=\varphi^{\prime }}$$

Finally, the RCS of the tag is calculated as:(25)$$\sigma_{\mathrm{tag}}\left(f\right)=\frac{Y_{tag}\left(f\right)}{\sum_{i=1}^{M}S_{i}\left(f\right)}$$
where $Y_{tag}\left(f\right)$ and $S_{i}\left(f\right)$ are the frequency-domain signals of $y_{tag}\left(t\right)$ and $s_{i}\left(t\right)$, respectively. For frequency-domain-coded chipless tags, the data can be obtained using Equation (25).

## 4. Simulation and Discussion

### 4.1. Performance Analysis

#### 4.1.1. SNR

Let the power corresponding to the signal $s_{i}\left(t\right)$ sent by the ith antenna be $P_{i}$. The received noise, $v_{m}\left(t\right)$, is Gaussian white noise with a mean of 0 and a variance of $\sigma^{2}$. The SNR of a single receiving antenna is as follows:(26)$$SNR_{siso}=\xi \frac{P_{i}}{\sigma^{2}}$$

In the case of using MIMO, the noise power in Equation (22) is as follows:(27)$$\begin{array}{c} E\left\{{\left|\overset{M}{\underset{m=1}{\sum }}v_{bm}\left(t\right)e^{-j\left(m-1\right)\varphi_{b}}\right|}^{2}\right\}\\ \;=E\left\{\left[\overset{M}{\underset{m=1}{\sum }}v_{bm}\left(t\right)e^{-j\left(m-1\right)\varphi_{b}}\right]{\left[\overset{M}{\underset{l=1}{\sum }}v_{bl}\left(t\right)e^{-j\left(l-1\right)\varphi_{b}}\right]}^{*}\right\}\\ \;=\overset{M}{\underset{m=1}{\sum }}\overset{M}{\underset{l=1}{\sum }}E\left\{v_{bm}\left(t\right)v_{bl}^{*}\left(t\right)\right\}e^{-j\left(l-1\right)\varphi_{b}}e^{j\left(m-1\right)\varphi_{b}} \end{array}$$

Because the noise received by different antennas is statistically independent, and the noise is white noise, we have the following:(28)$$E\left\{v_{bm}\left(t\right)v_{bl}^{*}\left(t\right)\right\}=\sigma^{2}\delta \left(k-l\right)$$
(29)$$E\left\{{\left|\sum_{m=1}^{M}v_{bm}\left(t\right)e^{-j\left(m-1\right)\varphi_{b}}\right|}^{2}\right\}=M\sigma^{2}$$

The signal power in Equation (22) is as follows:(30)$$\xi {\left\{\frac{\sin\left[\frac{M}{2}\left(\varphi_{f}-\varphi_{b}\right)\right]}{\sin\left[\frac{1}{2}\left(\varphi_{f}-\varphi_{b}\right)\right]}e^{j\frac{M-1}{2}\left(\varphi_{f}-\varphi_{b}\right)}\right\}}^{2}\overset{M}{\underset{m=1}{\sum }}s_{m}\left(t-\tau_{r}\right)$$

When the beam direction is consistent with the position of the tag, the signal power is $\xi {\left(M^{2}\right)}^{2}\sum_{i=1}^{M}P_{i}$. The transmitting power of all antennas is equal, so the SNR of this algorithm is as follows:(31)$$SNR_{mimo}=\xi M^{5}\frac{P_{i}}{\sigma^{2}}$$

Compared with SISO, the SNR is improved by $M^{5}$ times after MIMO signal processing. In addition, in the case of MIMO, the transmitting signal power is M times larger than that of a single antenna, so the actual SNR increases to $M^{4}$ times.

#### 4.1.2. Azimuth Resolution

For a radar antenna with aperture D, the half-power point width of its beam is [31] as follows:(32)$$BW_{HP}=0.886\frac{\lambda }{D}$$

For a linear array, the half-power beam width is as follows:(33)$$BW_{HP}=0.886\frac{\lambda }{\left(M-1\right)d}$$
where M is the number of antenna, d is the array element spacing, and λ is the wavelength of the electromagnetic wave. It can be seen that the aperture of the array antenna is larger than that for SISO, and the width of the antenna beam is smaller than that for SISO. The azimuth resolution of MIMO is, thus, improved.

### 4.2. Simulation

Figure 5 shows the simulation scenario.

The MIMO antenna array is a linear array, and the center of the array is located at the origin of the coordinates. The simulation parameters are given in Table 1. The number of array elements is M, and the array is arranged equidistant along the *X*-axis with half-wavelength intervals. The distance between the chipless RFID tag and the antenna array is L. The line from the tag to the origin has angle φ with respect to the *Y*-axis. The signal sent by the MIMO radar is a linear frequency-modulated continuous wave. The center frequency is $f_{c}$ and the bandwidth is B. The LFM signals sent by the MIMO radar are shown in Figure 6.

#### 4.2.1. Single-Tag Test

The proposed method was applied to the case where a signal tag is located in the reader zone. A 10-bit chipless RFID tag was constructed using FEKO software. The length of the tag was 55 mm and its width was 22 mm. It had 10 U-shaped resonators. Its structure and RCS are shown in Figure 7.

The green part in Figure 7a represents the dielectric material, and the yellow part is a perfect electric conductor. The proposed algorithm was used to estimate the orientation of the tag, and the target orientation diagram is shown in Figure 8a.

The energy of y(t) in (22) reached a maximum value in the 0° direction, so the tag is located in the 0° direction. The RCS is shown in Figure 8b. It can be seen that the algorithm was able to accurately estimate the tag azimuthal direction and compute the tag RCS.

#### 4.2.2. Noise Suppression Ability Test

The proposed method was applied in different SNR conditions. The test results are shown in Figure 9.

There are two curves in each figure. One is the real RCS based on FEKO software, and the other is the RCS obtained by the proposed method in the simulation environment. The test results indicate that this algorithm was able to measure the RCS accurately when the SNR was 0 dB.

#### 4.2.3. Comparison and Analysis

As a comparison, the RCS values obtained by a traditional single antenna under 0 dB, −10 dB, and −20 dB SNR conditions are shown in Figure 10. To ensure a fair comparison between the MIMO method and the SISO method, the transmitted power was normalized.

To compare the two methods quantitatively, we calculated the variance of the results.

From the comparison (Table 2), we can draw the conclusion that the RCS values obtained using the proposed method under the condition of −20 dB are better than those obtained using the traditional SISO method under the condition of −10 dB. The proposed algorithm can, thus, improve SNR suppression by 10 dB.

#### 4.2.4. Multi-Tag Test

Two of the same tags were placed 3 m away from the MIMO antenna, and the distance between them was d, as shown in Figure 11.

The value of d was set to 0.84 m, 0.86 m, 0.88 m, and 0.9 m in the configurations shown in Figure 11. The results of azimuth estimation are shown in Figure 12. When the distance between the two tags was 0.88 m, two peaks appeared in the direction diagram, and the orientation of the two tags was confirmed. It should be clarified that when the tags are on the same side, the power of the tag on the outer side is lower due to the beamforming characteristics of the linear antenna array. This problem can be avoided by using a circular array.

In the multi-tag recognition test, the tag used in the single-target test was regarded as Tag 1. The fifth resonator of Tag 1 was removed to construct a new tag, called Tag 2. Tag 2 is shown in Figure 13.

Figure 14a presents the multi-tag identification test scenario. Successful identification of the two tags is shown in Figure 14b,c.

## 5. Conclusions

In this paper, a signal processing algorithm for chipless RIFD multi-tag identification was proposed. Based on the mathematical descriptions presented in Section 2, a MIMO signal processing method was introduced for multiple chipless tag identification in Section 3. With the help of the space–time diversity characteristics of the MIMO antenna, the SNR was effectively improved, and the RCS could be calculated based on the azimuth location estimation. Two 10-bit U-shaped chipless RFID tags were constructed using FEKO software. The simulation results validated the theoretical explanations with good accuracy. It was shown that the proposed algorithm can achieve chipless RIFD tag reading at 3 m and can achieve a resolution of 0.88 m in the azimuth direction. A significant advantage of the method is its strong noise suppression. However, in order to simplify the algorithm, some approximations were made that caused performance degradation. By optimizing the algorithm, the performance can be further improved.

## Figures and Tables

**Figure 1 sensors-21-08314-f001:**
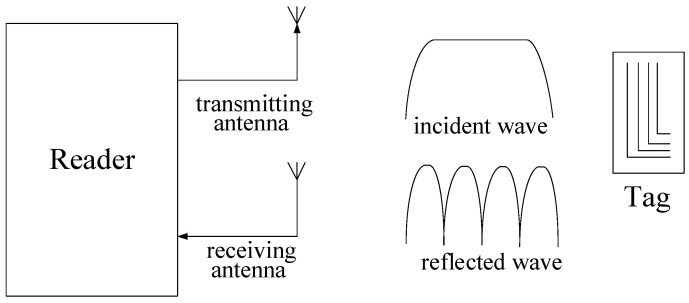
Principle of chipless RFID tags based on the frequency domain.

**Figure 2 sensors-21-08314-f002:**
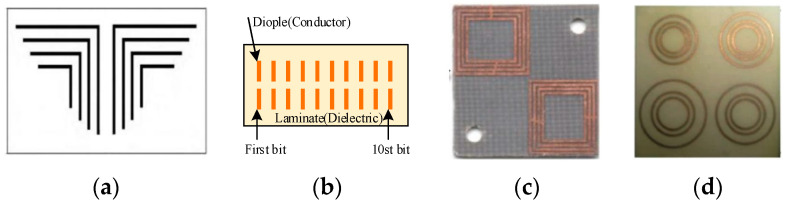
(**a**) Printable chipless tag; (**b**) Dipole array chipless tag; (**c**) Square ring chipless tag; (**d**) Ring-structure chipless tag.

**Figure 3 sensors-21-08314-f003:**
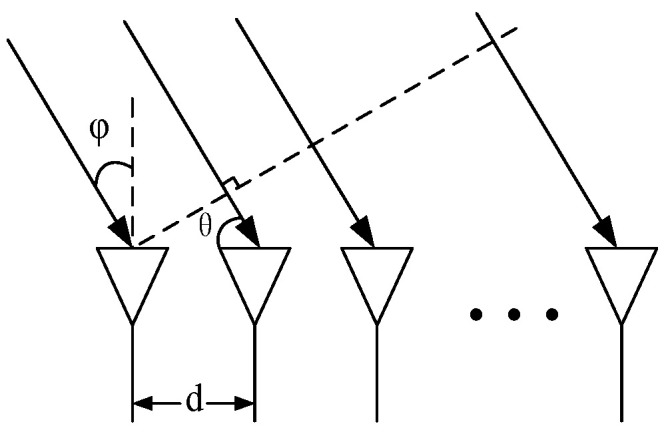
Uniform linear array structure.

**Figure 4 sensors-21-08314-f004:**
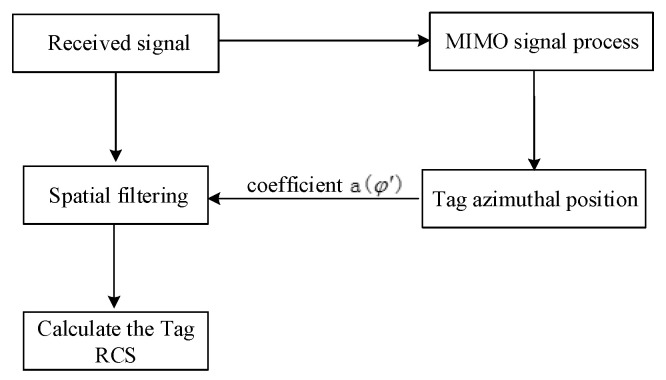
Chipless tag recognition algorithm flowchart.

**Figure 5 sensors-21-08314-f005:**
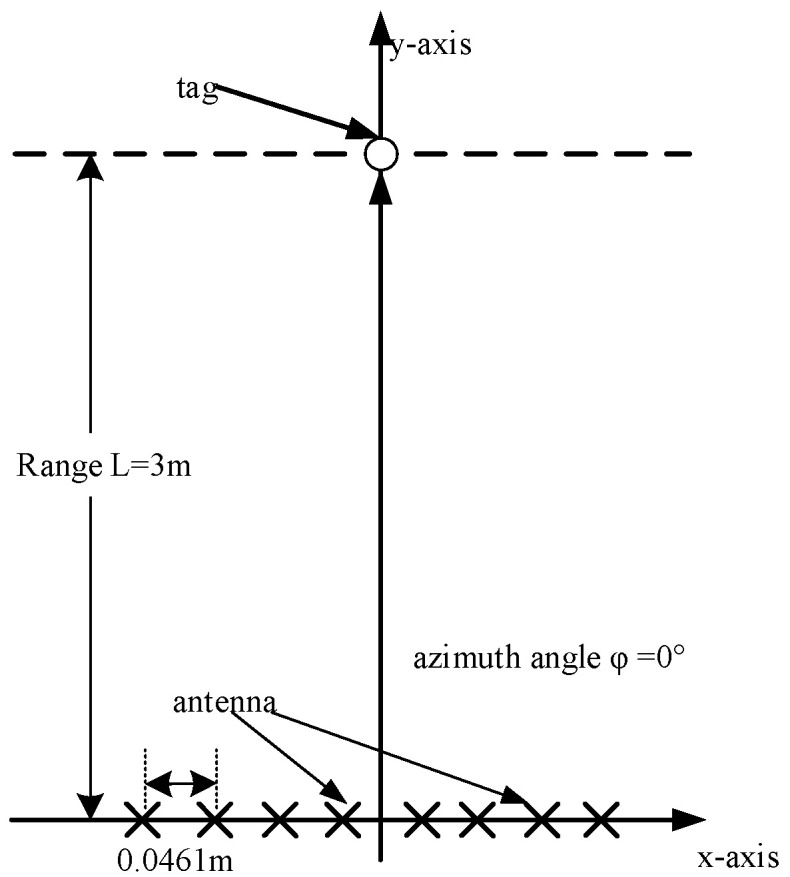
System configuration for a single-tag test.

**Figure 6 sensors-21-08314-f006:**
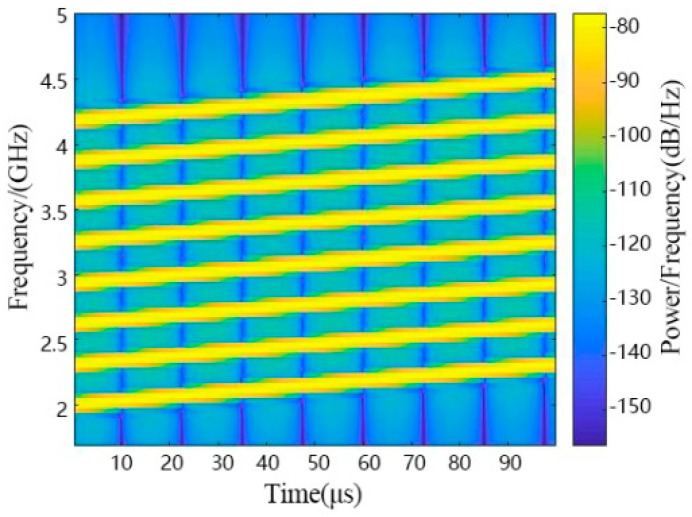
LFM signals sent by MIMO radar coding.

**Figure 7 sensors-21-08314-f007:**
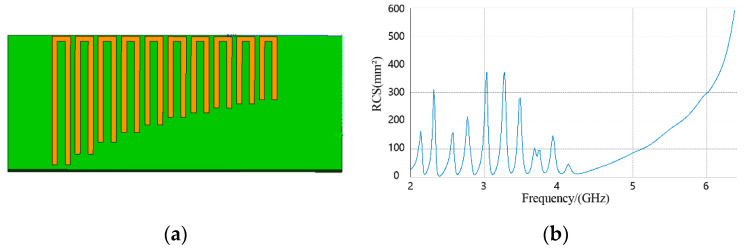
(**a**) Tag structure; (**b**) Tag RCS.

**Figure 8 sensors-21-08314-f008:**
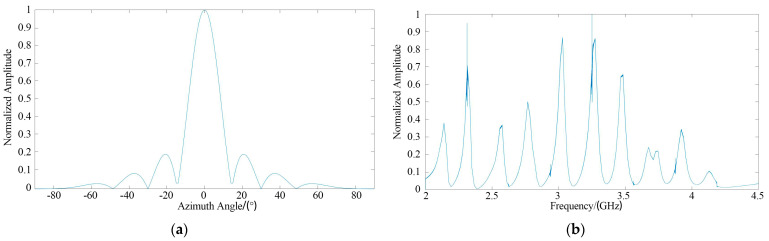
Single−tag test result: (**a**) Azimuthal direction estimation; (**b**) RCS.

**Figure 9 sensors-21-08314-f009:**
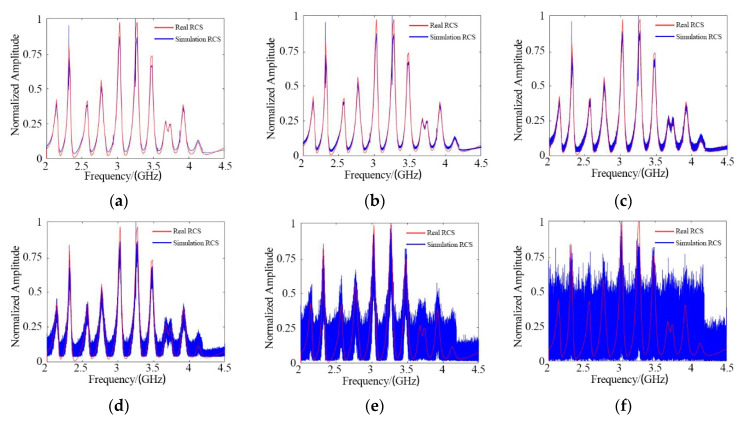
RCS results under different SNR conditions: (**a**) SNR = 40 dB; (**b**) SNR = 20 dB; (**c**) SNR = 10 dB; (**d**) SNR = 0 dB; (**e**) SNR = −10 dB; (**f**) SNR = −20 dB.

**Figure 10 sensors-21-08314-f010:**
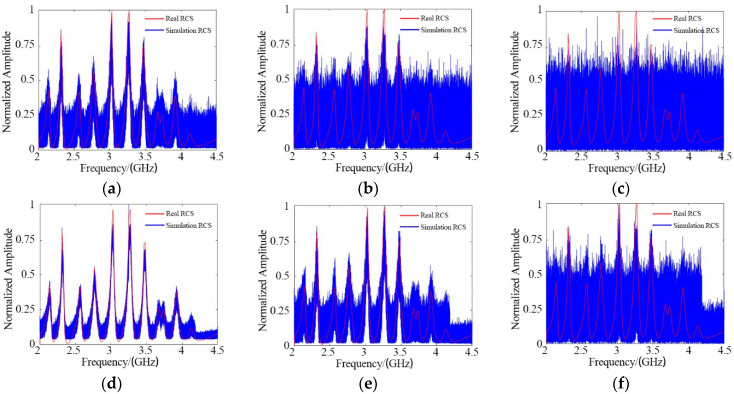
Comparison between the MIMO method and the traditional SISO method: (**a**) SISO, SNR = 0 dB; (**b**) SISO, SNR = −10 dB; (**c**) SISO, SNR = −20 dB; (**d**) MIMO, SNR = 0 dB; (**e**) MIMO, SNR = −10 dB; (**f**) MIMO, SNR = −20 dB.

**Figure 11 sensors-21-08314-f011:**
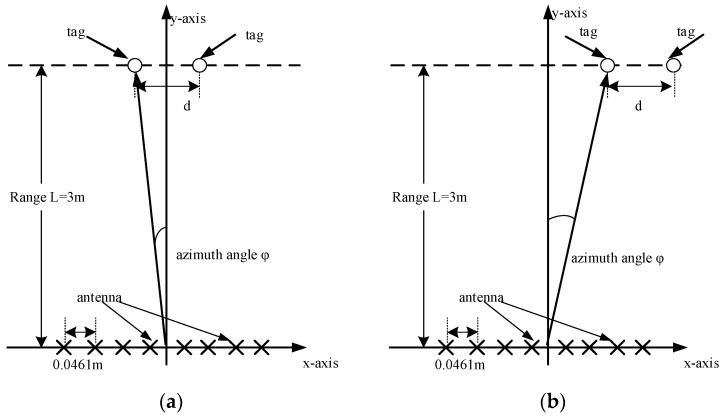
System configurations for the multi-tag test: (**a**) Tags placed symmetrically on the *Y*-axis; (**b**) Tags placed on the right of the *Y*-axis, $\varphi =5^{{}^{\circ}}$.

**Figure 12 sensors-21-08314-f012:**
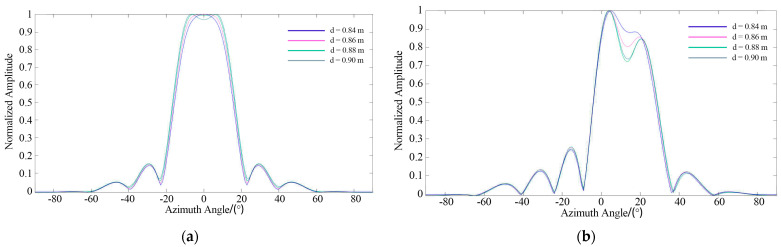
Azimuth estimation: (**a**) Symmetrical placement; (**b**) Same−side placement.

**Figure 13 sensors-21-08314-f013:**
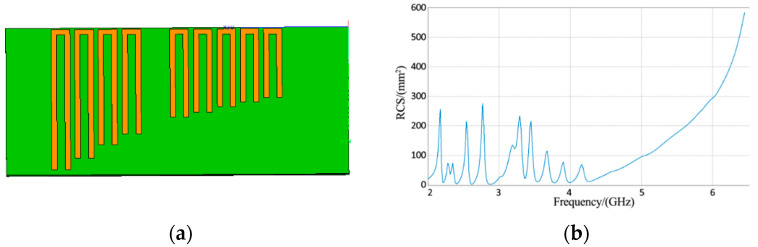
(**a**) Tag 2 structure; (**b**) Tag 2 RCS.

**Figure 14 sensors-21-08314-f014:**
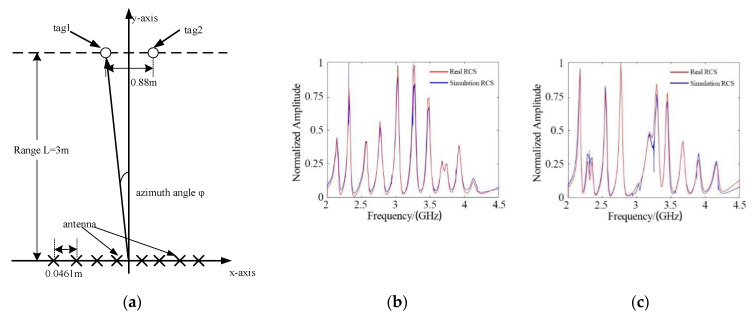
Multi-tag identification test: (**a**) Test Scenario; (**b**) Tag 1 RCS; (**c**) Tag 2 RCS.

**Table 1 sensors-21-08314-t001:** Simulation parameters.

Symbol	Value	Unit	Description
M	8	-	Number of antenna array elements
$f_{c}$	3.25	GHz	Center frequency
λ	0.0922	m	Wavelength
B	2.5	GHz	Bandwidth
$T_{r}$	100	μs	Pulse width
L	3	m	Distance between tag and MIMO radar
φ	0, 5	degree	Azimuth angle of the tag to the MIMO radar reference point
$BW_{HP}$	0.25314	rad	MIMO radar main beam width
δ	0.763	m	Azimuthal resolution

The symbol “-” means eight antennas in this MIMO system.

**Table 2 sensors-21-08314-t002:** Quantitative comparison.

SNR	MIMO Method Results Variance	SISO Method Results Variance
0 dB	$32\times 10^{-4}$	$93\times 10^{-4}$
−10 dB	$102\times 10^{-4}$	$399\times 10^{-4}$
−20 dB	$344\times 10^{-4}$	$533\times 10^{-4}$

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
