# Peer review of "A MIMO Radar Signal Processing Algorithm for Identifying Chipless RFID Tags"

_sensors, 2021, doi:10.3390/s21248314_

Round 1

Reviewer 1 Report

In this paper, the multiple-input-multiple-output (MIMO) radar signal processing algorithm is employed as an anti-collision methodology for the identification of multiple chipless radio-frequency identification (RFID) tags. In general, this paper lacks of comparison, which makes the presented method unconvincing. The authors should further enhance their experiments.

  1. In section 4.2, the authors aimed to validate their method based on simulations. However, the reviewer cannot find the performance comparison between authors’ method and traditional method. At this point, the reviewer or readers cannot know what the difference between authors’ method and traditional method.

  1. The authors just presented some figures in section 4. However, there are not quantitative parameters to evaluate the performance of presented method. At this point, the quantitative parameters must be discussed in detail to evaluate the presented method.

  1. The authors discussed the related works in section 2. However, the difference between authors’ method and these related works is not highlighted. The authors should clearly clarify this difference.

4. The English in this paper should be improved.

Author Response

Thank you for your valuable comments. We responded to all comments and improved the article according to the comments.

Reviewer 2 Report

The topic is very attractive. The idea of applying MIMO signal processing for enhancing the detection of the chipless RFID is wonderful. 

Threre are some critical points should be covered and improved by the authros:

  • The authors have claimed in the introduction that there is no protocols or MAC layer for chipless RFID systems for multitag detection. There are several contributions for chipless RFID systems considering the MAC protocol for multi-tag detection. The authors have to consider them in their literature review. 
  • The authors have considered that the RCS depending on the tag shape and its response (Frequency Selectivity), they have completely ignored the orientation of the tag and therefore they have used only the boresight scenario, which cannot clearly present the advantage of using their MIMO algorith of detecting the Tag Azimuthal Position at Zeor Degree. 
  • Detection of two tags, it was a special condition scenario of symmetric position to the Y-Access. Please try to use more general scenario without taking this special condition of symmetry to illustrate the robustniss of your system and reliablity of your results.
  • Using the center frequency of 4 GHz for the whole BW from 1 to 7 GHz for determing the wavelength and the corresponding distance between the antenna element is rather a tough approximation should be verified and explained.
  • For discussing the SNR improvements, please clarify that the total transmitted power is normalized to have a fair comparison with the single antenna reader.
  • Please explain how did you get the RCS in FEKO software [Monostatic or Bi-Static]. It is farfield probe based or total RCS effective area in [sqm].
  • In Fig 1: Incident wave in the diagram should be frequency flat since the authors didn't mention that they are using MB-OFDM.
  • Page 4: Line 125:"By measuring the power of the received signal and compensating the gain of the antenna,  the reader can obtain the RCS of the chipless RFID tag."      ...... What about the tag orientation here ?
  • Page 4: Line 125: "... frequency shift coding (FSK)" .... should be: "... frequency shift keying (FSK)"
  • Figures 8, 9, 10, ...  "Measured RCS compare to real RCS" .... You mean here the computed by analysis not the real measurements because this might lead to missunderstanding (Comparison between Measured RCS and Real RCS) .... Please try to illustrate it in better way to avoid this misunderstanding.
  • Page 9: Line 231:  "where  signal  tag  are  located  in  ... " should be:  where  a single  tag  is  located  in  ...
  • Page 9: Line 234: "in the figure 6. " should be:  "in figure 7 "

Genrally, the paper has a very good contribution but the above mentioned points should be considered and clarified.

Author Response

Thank you for your valuable comments.  We responded to all comments and improved the article according to the comments.   Please see the attachment.

Reviewer 3 Report

The authors have raised the interesting topic of decoding the multiple chipless RFID. This is the existing problem of all chipless RFID systems and it has not yet been thoroughly investigated. However, the paper does not clearly explain the issue and a critical assumption is violated (not satisfied). Please see below for detailed comments.     
1.    Grammar is needed to improve 
2.    Typos are in equations for example in (7), (8).
3.    Explanation of parameters in equations is not fully provided.
4. Equations 7 and 8 are the same, so do the authors mean transmitted signals and received signals are identical. 
5.    8 antennas array spacing d(half wavelength), so it makes up a synthesized length of 0.0375m*8 = 0.3 m. The reading distance is 0.5 m, so the assumption of signals arriving at the array is totally not satisfied. Signals are not parallel.     
6.    All equations are not cited or if authors proposed, they need to be proved.

Author Response

Thank you for your valuable comments. We have responded to all comments and improved the article according to the comments.

Round 2

Reviewer 1 Report

After revision, the paper can be accepted.

Reviewer 2 Report

The authors have improved the manuscript significantly and they have considered the comments.